# Factors Influencing the Implementation of Antimicrobial Stewardship in Primary Care: A Narrative Review

**DOI:** 10.3390/antibiotics12010030

**Published:** 2022-12-24

**Authors:** Véronique Suttels, Mathias Van Singer, Lauren Catherine Clack, Catherine Plüss-Suard, Anne Niquille, Yolanda Mueller, Noémie Boillat Blanco

**Affiliations:** 1Infectious Diseases Service, Lausanne University Hospital and University of Lausanne, 1011 Lausanne, Switzerland; 2Institute for Implementation Science in Health Care, Medical Faculty, University of Zurich, 8006 Zürich, Switzerland; 3Swiss Centre for Antibiotic Resistance, Institute for Infectious Diseases, University of Bern, 3001 Bern, Switzerland; 4Center for Primary Care and Public Health (Unisanté), Pharmacy University of Lausanne, 1011 Lausanne, Switzerland; 5Institute of Pharmaceutical Sciences of Western Switzerland, University of Lausanne, 1011 Lausanne, Switzerland; 6Center for Primary Care and Public Health (Unisanté), Department of Family Medicine, 1011 Lausanne, Switzerland

**Keywords:** antimicrobial stewardship, antimicrobial resistance, primary care, qualitative

## Abstract

Antimicrobial resistance (AMR) is directly driven by inappropriate use of antibiotics. Although the majority of antibiotics (an estimated 80%) are consumed in primary care settings, antimicrobial stewardship (AMS) activities in primary care remain underdeveloped and factors influencing their implementation are poorly understood. This can result in promising stewardship activities having little-to-no real-world impact. With this narrative review, we aim to identify and summarize peer-reviewed literature reporting on (1) the nature and impact of AMS interventions in primary care and (2) the individual and contextual factors influencing their implementation. Reported activities included AMS at different contextual levels (individual, collective and policy). AMS activities being often combined, it is difficult to evaluate them as stand-alone interventions. While some important individual and contextual factors were reported (difficulty to reach physicians leading to a low uptake of interventions, tight workflow of physicians requiring implementation of flexible and brief interventions and AMS as a unique opportunity to strengthen physician-patients relationship), this review identified a paucity of information in the literature about the factors that support or hinder implementation of AMS in primary care settings. In conclusion, identifying multilevel barriers and facilitators for AMS uptake is an essential step to explore before implementing primary care AMS interventions.

## 1. Introduction

Antimicrobial resistance (AMR) is directly driven by inappropriate use of antibiotics [1,2]. Antibiotic consumption in the outpatient setting represents 80% or more of total antibiotic consumption in Europe and the United States [3,4,5,6,7]. Inappropriate prescribing accounts for up to fifty per cent of all antibiotic consumption in outpatient care (including unnecessary prescription or inappropriate selection, dosing and duration of treatment) [8]. Inappropriate use of antibiotics is common for the most frequent infections usually managed by primary care physicians such as upper and lower respiratory tract infections (RTI) and urinary tract infections (UTI) [9,10,11].

Antimicrobial stewardship (AMS) programs, defined as “coordinated interventions designed to improve and measure the appropriate use of antibiotics”, are developed to tackle the challenge of over-consumption of antibiotics [12]. AMS remains the cornerstone of the global effort to slow the spread of antimicrobial resistance and maximize the benefits of antibiotic treatment [13]. Such programs are common practice in acute care settings, such as hospitals, while AMS remains virtually absent in outpatient practice [14,15]. This setting presents a challenge to AMS efforts, in part, because of its partner heterogeneity (private practices with varying numbers of general practitioners (GPs) and specialists; public and private outpatient clinics) and important geographic spread [13]. While some AMS interventions show promising and effective results in clinical trials, there is little evidence that such interventions make their way into standard practice, leading to frustratingly low real-world impact. To bridge the gap between effectiveness and successful implementation, understanding of the individual and contextual factors affecting antibiotic prescribing and uptake of AMS activities in primary care is primordial and listed as a key priority in AMS research [16]. Understanding such factors is critical to designing implementation processes tailored to the primary care setting. Considering that the common indications for which antibiotics are prescribed are usually managed by primary care physicians, this is the focus of the current review.

With this narrative review, we aimed to summarize peer-reviewed literature reporting on (1) the nature and impact of AMS activities targeting physicians in primary care and (2) the specific behavioral and contextual factors that influence the uptake of AMS activities in primary care.

## 2. Methods

We searched for peer-reviewed literature in PubMed and Google scholar databases published prior to October 2022. We considered reviews and primary studies published since 2000 in English. Search terms included “Antimicrobial stewardship”, “AMS”, “primary care”, “qualitative”. Regarding selection criteria, we included clinical trials, reviews and meta-analyses that reported AMS interventions in primary care and nursing homes and/or observational quantitative and qualitative studies, clinical trials, reviews and meta-analyses that evaluated factors influencing the implementation of AMS in the community (primary care, nursing homes and pharmacies). We excluded studies which targeted a different setting, such as emergency department or hospital. References of included studies were also screened to identify further relevant articles meeting the selection criteria (having inclusion criteria and not having exclusion criteria) of this narrative review. For included articles, reported AMS activities were identified and categorized according to different contextual levels: (1) Individual level behavior change; (2) Collective (team, organization) level change, and (3) Structural/policy/legal level change as suggested by the Consolidated Framework for Implementation Research (CFIR) [17].

## 3. Results

In this narrative review, we discuss 30 studies reporting on factors influencing the implementation of antimicrobial stewardship in primary care. The characteristics of the studies are shown in Table 1 and a summary of the findings in Table 2.

### 3.1. Reviews on AMS Activities in Primary Care

A multitude of AMS activities targeting physicians, patients in primary care and/or the public have been reported [18]. A 2014 meta-analysis including 50 studies on AMS interventions in primary care identified that programs including communication skills training and laboratory testing were associated with a significant reduction in antimicrobial consumption. Evidence behind the impact of other stewardships activities was lower [19].

To guide implementation of the broad range of AMS intervention possibilities, the CDC proposed a framework of four core components targeting different contextual levels in 2016: commitment, action for policy, tracking & reporting and education & expertise [13].

In the following paragraphs, we will briefly review the impact of different AMS interventions and detail the factors influencing their implementation by based on the different contextual levels previously defined.

### 3.2. Individual Level Behavior Change

#### 3.2.1. Education of Patients

Education for patients can be provided passively by the clinician with the help of leaflets and/or more actively in conjunction with education of prescribers on communication skills, which can enable shared decision-making and explore the patients’ values and preferences [20]. Visual decision-making aids are also an attractive candidate for patients’ education on AMS in primary care [21]. These easy-to-understand infographics on the benefits, and harms to expect of a certain treatment or screening test have already been evaluated for cancer screening improving patients’ informed decision making and reducing the number of screening tests [22].

Educational interventions targeting patients are a key component in AMS but there is a lack of evidence demonstrating their impact as a stand-alone intervention [23].

According to a qualitative study conducted in France, primary care prescribers believe that patients’ education is part of their duty [24]. Depending on the setting, between 10 to 75% of patients reported the desire to receive antibiotic treatment [25]. This desire to receive antibiotics may affect their satisfaction, especially in the absence of information, reassurance and a clear plan from the physician [25,26]. However, the perception of patients’ desire by the GP, rather than patients’ actual desire, is strongly associated with inappropriate antibiotic consumption highlighting the need to also provide education to GPs [25,26]. Pharmacists also provide education for patients by counseling them about the appropriate use of antibiotics, improving adherence to antibiotic prescriptions (including delayed ones) or decreasing patient perception of the need for an antibiotic in viral illnesses when they are not indicated, such as in acute RTIs [27].

#### 3.2.2. Education of General Practitioners

Several formats of educational sessions target physicians in primary care: lectures part of continuing medical education, online interventions and interprofessional team discussions [23]. In a similar way to the education of patients, stewardship activities usually include education of GPs as a reinforcement of other interventions, which makes it difficult to evaluate the impact of GPs education itself on antibiotic prescribing practices.

Recently, the nationwide implementation of an online communication skills training illustrated the difficulty of reaching high participation rate in a real-life setting. In this randomized trial, only 3% and 1% of GPs completed the first and second educational trainings, which was the main reason explaining the lack of impact of the intervention on antibiotic prescription [28].

A recent qualitative study identified factors affecting the uptake of educational interventions by GPs: first, facilitators: (1) having flexible and relevant learning strategies; (2) having easy to access information, resources and reminders; and (3) creating a heightened awareness about AMR; and, second, barriers: time pressure (mostly linked to patient volume) [15].

#### 3.2.3. Communication Skills Training

Training physicians in communication skills facilitates shared decision making with patients by clarifying misperceptions and presenting available evidence behind benefits and harms of treatment. A meta-analysis showed moderate quality evidence that interventions aiming to facilitate shared decision making and help physicians communicate with patients reduce antibiotic use for acute RTI in primary care [29]. Communication skills training as a stand-alone intervention seems to reduce antibiotic consumption under trial conditions [30]. Most often though, this strategy is combined with other interventions, such as point-of-care laboratory testing. However, even when acquired, time pressure is the main barrier to the use of communication skills as GPs mention that it takes more time to convince patients they have a viral infection and do not need antibiotics rather than make a prescription [15]. Shorter consultation time results in less reassurance and information given about the lack of utility of antibiotics [15,31,32]. As stated in the previous paragraph, another important barrier to communication skills training is the uptake of training by physicians [28]. Other barriers identified in a qualitative study for the optimization of the management of UTIs in primary care via shared decision making included misunderstanding of depth of knowledge between GP and patient, miscommunication between the patient and the GP and the nature of the consultations (i.e., phone consultation) [33].

#### 3.2.4. Delayed Prescribing

Delayed prescribing can be a useful strategy particularly when a patient expects to receive an antibiotic. It is a safe strategy which lead to a reduction in antibiotic use in RTI consultations even in high-risk patients compared to immediate antibiotics [34,35]. Interestingly, delayed prescribing does not affect patient satisfaction compared to immediate antibiotics, as this strategy gives both the GP and patient a sense of security [35]. Perceived as a safety net in case of diagnostic uncertainty or logistic restraints, GPs also view the delayed prescribing as educational and empowering to patients, strengthening the patient-physician relationship even [36]. It further helps patients to avoid after-hours consultation. On the other hand, it can sometimes be experienced as a loss of control over management decisions and is not suitable for patients who are not able to understand the indications for antibiotics [36,37]. GPs note that a factor supporting the use of delayed prescribing was prior experience with this strategy. GPs also mentioned the need of greater uniformity within and between practices on delayed prescribing, which could be reinforced by training, guidelines and feedback [38].

#### 3.2.5. Electronic Clinical Decision Support Tools

Computer-assisted clinical decision support tools (CDSS) provide the prescribers an easy and rapid access to information and assist GPs at the point of prescribing. It can do so by creating alerts, proposing antibiotic order sets or by prompting questions to guide the choice, dosage and duration of treatment in the patient’s electronic health record.

Under trial conditions and mostly for RTIs in primary care, CDSS induce a marginal to moderate reduction in antibiotic prescribing [39]. Very few interventions report pre-deployment stakeholder analyses or prescriber decision mapping to justify the intervention design [40]. Barriers to implementation are interruption of the workflow causing additional time pressure, and inflexibility of the application [41]. Besides these practicalities, the quality and perceived usefulness of the tool’s content to assist local AMS also seem an essential feature [42].

#### 3.2.6. Biomarkers at the Point-of-Care: C Reactive Protein and Procalcitonin

Biomarkers of inflammation are elevated in the acute phase response to tissue injury irrespective of its etiology. They can safely guide clinicians prescribing decision by ruling out severe bacterial infections. A recent Cochrane review including 12 trials concludes that point-of-care C reactive protein (POC-CRP) safely reduces antibiotic prescription among patients with acute RTI in primary care [43]. There were differences between studies regarding the CRP cut-off values applied to guide antibiotics. In some studies, the recommendation was rather vague, while, in other studies, the recommendation was based on different numeric cut-offs (≥40 mg/L; >50 mg/L; >60 mg/L; ≥100 mg/L for an immediate prescription of antibiotics). A Swiss cluster-randomized study also showed a decrease in antibiotic prescription among patients with lower RTIs in primary care with the use of point-of-care procalcitonin (POC-PCT) at a cut-off of ≥0.25 µg/L compared to usual care [44]. POC-CRP and -PCT are attractive tools as they specifically address diagnostic uncertainty, a known barrier to appropriate antibiotic prescription [15,45,46,47], while the risk of refraining from prescribing antibiotics (e.g., complications) is generally overestimated by the GP [48]. However, the long-lasting effect of POC-CRP was recently questioned when analyzing the impact 12 months after its implementation. Indeed, the early improvement seen with CRP disappeared with time mainly due to reduced use. The time required to perform the test might be a barrier to long-term engagement [49]. However, a qualitative study in high prescribing practices in England about the implementation of POC-CRP guided prescriptions (and delayed prescription) found that GPs deemed this strategy had a limited value as clinical tool, and was useful only in rare instances of clinical uncertainty and/or for clinicians less experienced. However, it was seen as a helpful “social tool” to negotiate treatment while maintaining GP-patient relationships or educating patients [50].

A nationwide prospective web-based survey evaluated GP attitudes towards POC-CRP. It showed that GPs would use lower CRP cut-offs to guide prescribing for more severe RTIs than for uncomplicated RTIs. Faced with intermediate CRP results in non-severe patients, GPs preferred to postpone decision on antibiotic prescription by 3 to 5 days rather than to write a delayed prescription [45]. For both POC-CRP and -PCT, GP’s attitudes are mostly positive, as they feel it allows for safe reduction of antibiotic consumption for RTIs. Reimbursement issues, the need for quality control and the negative impact on work flows were other factors affecting the adoption of POC-CRP and -PCT in primary care [51,52].

In a study analyzing factors associated with overruling of POC-PCT guidance in the setting of a clinical trial, some GPs characteristics (GP’s number of years of experience [median of eight year among those who did not overrule PCT guidance versus 10 years among those who overruled PCT guidance] as well as GPs working in an urban setting) were associated with antibiotic prescription in spite of low PCT levels, highlighting the general behavioral problem of overprescription by physicians. (Knüsli 2022—unpublished data, [10]).

### 3.3. Collective (Team, Organization) Level Change

#### 3.3.1. Guideline Dissemination

National and international guidelines exist for the most common infections encountered in primary care. They are mostly developed by professional organizations (for example national infectious diseases societies) or, less often, publicly funded national institutes. The dissemination of guidelines is classically a top-down intervention and is often part of multifaceted interventions [20]. When used as a stand-alone intervention, mixed results were obtained with no effect to a modest overall decrease in antibiotic consumption accompanied by increased use of recommended antibiotics [11,53,54]. Although easily accessible guidelines are an appreciated resource by GPs and GPs are often aware of guidelines on common infections such as upper respiratory tract infections and of the preponderance of viral RTI in their setting, they tend to prescribe antibiotics because of concern of adverse patients outcome without antibiotics [55]. Data on the age of GPs are inconsistent, with some studies showing an association between older age and antibiotics prescription, others not [56]. Guidelines dissemination seems to be less effective to change prescribing of older GPs as demonstrated in the treatment of uncomplicated UTI, where the GP’s age was directly associated with prescribing antibiotic therapy not recommended by guidelines (e.g., fluoroquinolones) [57].

Older GPs might distrust guidelines in favor of their own clinical impression because of their extensive expertise or might have kept more liberal prescribing practices. Additionally, impact of guidelines dissemination might be lower among older GPs because of lower knowledge of current and updated guidelines as shown in the treatment of cardiovascular diseases [57,58].

#### 3.3.2. Multifaceted Intervention Deployed by a Large Health Care Organization

The impact, effectiveness, and safety of implementing the CDC core elements [13] as a whole was assessed in a quasi-experimental controlled study focusing on uncomplicated acute RTI. The study showed a safe but modest reduction in antibiotic consumption. A post-implementation survey indicated that while site champions were comfortable delivering the bundle intervention, there were time constraints preventing them from carrying out their tasks properly [59].

#### 3.3.3. Provider Feedback

Provider feedback is a top-down intervention, which does not give extra work to physicians. Usually, such interventions include thousands of physicians. Feedback on prescription habits is usually organized at the level of groups of physician practices sharing a common electronic health record, or by health insurers based on billing data. Personalized feedback interventions are often part of multifaceted interventions (e.g., guidelines diffusion, provider education, peer comparison). Mixed results were obtained, as some studies showed no reduction in antibiotic prescribing [60], while others showed a moderate reduction in antibiotic consumption, especially when designed with insights from behavioral sciences (e.g., peer comparison or accountable justification) [61,62,63,64]. Qualitative data suggests ‘deep skepticism’ about the usefulness of audit data on antimicrobial prescribing [65,66,67]. Physicians mention that some aspects of the feedback are presented in a complex manner which a difficult to process in a time-constrained environment. They would favor a visual presentation of the data [67]. In nursing homes, physician’s feel that audit reports did not impact their prescribing as they were already aware of the problem. They tend to welcome broader interventions where audit is complemented by education of nurses and communication skills training [68].

#### 3.3.4. Quality Circles

Quality circles (QCs) are made up of 6 to 12 primary healthcare professionals who regularly meet to reflect on and improve their standard practice. Different types of QCs exist, e.g., with or without pharmacists. In Switzerland, the interprofessional QCs approach has shown potential to reduce antibiotic prescribing by GPs in private practice and in nursing homes [69]. These physician-pharmacist quality circles are based on open exchange of experiences, new knowledge acquirement and implementation. Regarding antibiotic use, this approach is operationalized through: (1) the analysis of antibiotic prescription data of each GP (annual proportion of patients with at least a prescription, profile of molecules prescribed, etc.) incl. benchmarking within the group and with GPs non active in such QC; (2) the dissemination of good clinical practice recommendations according to evidence-based medicine; (3) the definition of a local drug treatment consensus per common infection with conservative use of antibiotics; if antibiotics are needed, the choice is based on the selection of the spectrum of activity, adverse effects, interaction profile, local resistance and package size adapted to treatment duration; (4) the application of the consensus by the GPs involved; (5) the continuous improvement through revision of the consensus every 1 to 2 years to integrate new evidence and discussion of the effective changes in practice [70,71]. QCs can improve standard practice like prescription patterns and diagnostic habits, enhance professional development and psychological well-being in GPs. However, the results of randomized controlled trials are inconsistent and offer only limited behavioral explanations for these positive effects [72]. In the study by Klepser et al. [73] community pharmacists used rapid POC tests to guide clinical decision making as appropriate under a physician-led evidence-based protocol to treat patients with influenza and group A Streptococcus pharyngitis. This model pairing physicians and community pharmacists led to a more prudent use of antibiotics while providing safe and convenient care for patients. A recent GP–pharmacist collaboration showed effective implementation of delayed prescribing, educational programs or by reviewing broad-spectrum antimicrobial prescribing [74]. Readiness is a key factor for changing antibiotic prescribing in primary care. The most notable distinguishing characteristics between the high and low readiness-to-change practices were with regard to the nature and quality of group dynamics including communication, learning climate and cohesion [75]. The importance of group dynamics motivated the conception of QCs in European primary care.

### 3.4. Structural/Policy/Legal Level Change

#### 3.4.1. Education of the Public

Public campaigns are done in many countries to provide information on the appropriate use of antibiotics in outpatients. There is a wide variation in the intensity and type of campaign, some based on paper leaflets or simple internet messages to wide and expensive mass-media campaigns [76]. Most campaigns target the public and the physicians at the same time. Most often, health authorities implement these campaigns as part of their national strategy to reduce antimicrobial resistance. All campaigns focus on respiratory tract infections as they are the reason for most prescriptions of antibiotics.

Most public campaigns seemed to reduce antibiotic use, but there is a lack of evidence demonstrating their impact as a stand-alone intervention [23,77,78]. Multifaceted informational campaigns coupled with GP and pharmacist education repeated over several years appear to have the greatest effect [76,79,80,81].

#### 3.4.2. Governmental Strategies

Many governments have national antimicrobial resistance actions plan, which can be very diverse. Seventeen government policy interventions have been described worldwide: most commonly public awareness campaign, guidelines, changing regulations around prescribing and reimbursement. Unfortunately, because of a lack of rigorous evaluations, their impact on antimicrobial use remains unclear. Most of these policies focus on changing the habits of physicians, rather than targeting other healthcare professionals or altering healthcare structures to reduce antibiotic consumption [82]. However, in Sweden, since the mid-1990s, the governments and health authorities took a bottom-up regulatory approach to the risks of AMR by establishing a program including benchmarking, locally adapted guidelines and restriction in reimbursement, complemented by public awareness campaigns with since then a gradual decrease in antibiotics consumption [83].

Stakeholders’ views on different government interventions are rarely described. Notably, a qualitative study in France indicated that GPs prefer government interventions not directly targeting prescribers. GPs rather preferred indirect interventions such as increasing the unit sales price of antibiotics, the restricted reporting of susceptibility tests, or the limitation of the number of molecules available in primary care [24].

Governments should implement specific rules, funding and legislation for the antibiotic dispensing circuit, specifically of unit-dose antibiotics, in community and hospital pharmacies [84].

## 4. Discussion

Overall, this narrative review aimed to identify factors influencing the uptake of AMS activities in primary care. Indeed, several studies showed a low uptake of such activities, which jeopardizes their real-life impact on antibiotics consumption. Several individual, collective and policy AMS activities are usually combined, which makes them difficult to evaluate as a stand-alone intervention. Although there is a paucity of qualitative research on the uptake of AMS interventions in primary care, we still identified some recurrent patterns affecting the implementation of AMS activities. First, the main barrier to successful implementation of most AMS activities is the difficulty to reach primary care physicians. It is mainly due to the heterogeneity of GPs and to time constraints in their daily work, which jeopardizes the uptake of AMs activities. Quality circles seems to be a promising setting to enhance the uptake of AMS activities in primary care. Second, the tight workflow of GPs highlights the need of having flexible, easy-to-access and brief AMS activities. Third, several AMS activities are a unique opportunity of strengthening the physician-patients relationship, mainly through communication skills training and laboratory testing which reduce diagnostic uncertainty and can be used as a facilitator of implementation.

## 5. Conclusions

We identified some elements affecting the implementation of AMS activities, main barriers being the difficulty to reach primary care physicians and the tight workflow of GPs, while a facilitator being the opportunity to strengthen physician-patients relationship.

Failure to take these factors into account can result in stewardship activities resulting in little-to-no real-world impact. A thorough understanding of the individual and contextual factors that drive current behaviors and influence implementation of AMS activities is necessary to inform the systematic, tailored design of approaches to implement AMS in primary care setting.

## Figures and Tables

**Table 1 antibiotics-12-00030-t001:** Characteristics of studies on factors influencing the implementation of antimicrobial stewardship activities and categorized according to different contextual levels.

Contextual Level	Type of Intervention	First Author Publication Year	Study Design	Location	Sample Size
Individual level change	Education of patients	O’Connor 2018	Narrative review	NA	NA
		McKay R 2016	Systematic Review	NA	NA
	Education of GPs	Jeffs 2020	Qualitative study	Canada	23 HWs
		D’Hulster 2022	Clinical trial	Belgium	10375 GPs
	Communication skills	Jeffs 2020	Qualitative study	Canada	23 HWs
		Fletcher-Lartey 2016	Bi-annual survey and qualitative study	Australia	32 GPs
		Kumar 2003	Qualitative study	United Kingdom	40 GPs
		D’Hulster 2022	Clinical trial	Belgium	10375 GPs
		Lecky 2020	Qualitative study	United Kingdom	20 GPs and 29 patients
	Delayed prescribing	Spurling 2017	Systematic Review	NA	
		Dallas 2020	Qualitative study	Australia	22 HWs
		Høye 2010	Qualitative study	Norway	33 GPs
		Ryves 2016	Qualitative study	England	32 GPs
	Electronic clinical decision support tools	Forest 2014	Narrative review	NA	NA
		Kortteisto 2012	Qualitative study	Finland	48 HWs
	Biomarkers at the point-of-care: C reactive protein and procalcitonin	Jeffs 2020	Qualitative study	Canada	23 HWs
		Martínez-González 2022	Web-based survey	Switzerland	188 GPs
		Lecky 2020	Web-based survey	United Kingdom	428 GPs
		Lopez-Vazquez 2011	Systematic Review	NA	NA
		Little 2019	Discussion of a clinical trial	NA	NA
		Borek 2021	Qualitative study	England	50 HWs
		Cals 2010	Qualitative study	Netherlands	20 GPs
		Geis 2022	Qualitative study	Switzerland	12 GPs
		Knusli 2022	Secondary analysis of a clinical trial	Switzerland	60 GPs
Collective level change	Guideline dissemination	Md Rezal 2015	Systematic Review	NA	NA
		Martinez-Gonzales 2020	Cross-sectional study	Switzerland	155’292 patients
		Plate 2020	Cross-sectional study	Switzerland	163 GPs practices and 1352 patients
		Hoorn 2019	Review	NA	NA
	Multifaceted intervention deployed by a large health care organization	Madaras-Kelly 2021	Post-implementation survey	USA	Unknown
	Provider feedback	Szymczak 2014	Qualitative study	USA	24 pediatricians
		Zetts 2020	Qualitative study	USA	26 GPs and 26 pediatricians
		Roche 2022	Qualitative study	Ireland	12 GPs
		Laur 2021	Qualitative study	Canada	18 GPs
	Quality circles	Elango 2018	Survey and Qualitative study	USA	31 HCWs
Policy level change	Education of the public	None			
	Governmental strategies	Mauffrey 2016	Qualitative study	France	30 HCWs

Abbreviations: NA: not applicable; HW: healthcare worker; GP: general practitioner.

**Table 2 antibiotics-12-00030-t002:** Summary of barriers and facilitators influencing the implementation of AMS activities.

Contextual Level	AMS Activity	Barriers	Facilitators
**Individual level behavior change**	*Education of patients*	Perceived patient desire to receive antibiotics	Part of GP duty Providing reassurance and a clear plan
	*Education of general practitioners*	Low participation Time pressure (linked to patient volume)	Flexible and relevant learning strategies Easy to access information, resources and reminders Creating a heightened awareness about AMR
	*Communication skills training*	Low participation Time pressure (short consultation time) Misunderstanding of depth of knowledge of the patient Phone consultations	
	*Delayed prescribing*	Loss of control over management decisions Less suitable for patients not fully understanding AB indications Heterogeneity between and within practices	High patient satisfaction Perceived as a safety net Educational and empowering to patients Improving patient-physician relationship Avoiding after-hours consultation
	*Electronic clinical decision support tools*	Interruption of workflow and additional time pressure Inflexibility of the application	High quality application Perception of content as relevant
	*Biomarkers at the point-of-care: C reactive protein and procalcitonin*	Reduced use over time Perceived as of limited clinical value	Addresses diagnostic uncertainty Helpful for unexperienced GPs “Social tool” to negotiate treatment and educate the patient
**Collective (team, organization) level change**	*Guideline dissemination*	Older GPs Concern of adverse patient outcome without ABs	Easy access
	*Multifaceted intervention deployed by a large health care organization*	Time pressure	Site champions being comfortable delivering the bundle intervention
	*Provider feedback*	Skepticism about the usefulness of audit data Complex reports to read No perceived impact on prescribing	Easy understandable visual data representation Feedback complemented by nurses education and communication skills training
	*Quality circles*		High readiness Positive group dynamic
**Structural/policy/legal level change**	*Education of the public*		Mass public campaigns coupled to GP education
	*Government strategies*	Directly targeting prescribers	Bottom-up approach and indirect interventions (e.g., local guidelines, reimbursement restrictions, restricted reporting of susceptibility tests, mass public campaigns)

## Data Availability

Not applicable.

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
