# Peer review of "Factors Influencing the Implementation of Antimicrobial Stewardship in Primary Care: A Narrative Review"

_antibiotics, 2022, doi:10.3390/antibiotics12010030_

Round 1

Reviewer 1 Report

This well-written narrative review covers essential aspects of implementing antimicrobial stewardship programs in primary care settings. However, the following issues need to be addressed before considering this manuscript for publication:

  • Line 40: “Europe” is misspelled here.
  • While the authors addressed essential factors affecting the implementation of antimicrobial stewardship in primary care settings, little was discussed about the role of pharmacists, especially infectious diseases clinical pharmacists, in implementing and maintaining the antimicrobial stewardship programs in these sittings, educating the primary care providers about the best practices of antibiotic prescribing, and counseling patients about the appropriate use of antibiotics. 
  • Line 321: The word “promising” might not be used correctly here.

Author Response

#1

This well-written narrative review covers essential aspects of implementing antimicrobial stewardship programs in primary care settings. However, the following issues need to be addressed before considering this manuscript for publication:

  • Line 40: “Europe” is misspelled here.

We corrected the misspelling.

  • While the authors addressed essential factors affecting the implementation of antimicrobial stewardship in primary care settings, little was discussed about the role of pharmacists, especially infectious diseases clinical pharmacists, in implementing and maintaining the antimicrobial stewardship programs in these sittings, educating the primary care providers about the best practices of antibiotic prescribing, and counseling patients about the appropriate use of antibiotics. 

We thank the reviewer for this important comment.

We added information on the role of pharmacists in implementing and maintaining antimicrobial stewardship programs, physicians education and patients counselling: in the paragraphs on “education of patients” and “quality circles”.

·        Line 321: The word “promising” might not be used correctly here.

We removed the word “promising”.

Reviewer 2 Report

Dear Author, 

Your article Factors influencing the implementation of antimicrobial stewardship in primary care: a narrative review is interesting, but should be improve: 

- please add information about inclusion criteria and exclusion for article to review

- References of included studies were also screened to identify further relevant articles - Please provide information What is relevant articles or not ? Improve design of review

My decision is accept after minor revision. 

Author Response

#2

Dear Author, 

Your article Factors influencing the implementation of antimicrobial stewardship in primary care: a narrative review is interesting, but should be improve: 

- please add information about inclusion criteria and exclusion for article to review

We thank the reviewer for this helpful comment which improved the methods section of the manuscript.

We provided more details on inclusion and exclusion criteria for articles included in the methods part of this narrative review.

We also included a table (Table 1) summarizing the characteristics of the studies included in our review.

- References of included studies were also screened to identify further relevant articles - Please provide information What is relevant articles or not ? Improve design of review

My decision is accept after minor revision.

We clarified this point in the methods part:

“References of included studies were also screened to identify further relevant articles meeting the selection criteria (having inclusion criteria and not having exclusion criteria) of this narrative review. “

Reviewer 3 Report

I think the work is very well done.

However, I allow myself to make two observations.

Missing reference in paragraph: line 124. A recent qualitative study identified factors affecting the uptake of educational interventions by GPs: first, facilitators: 1) having flexible and relevant learning strategies; 2) having easy to access information, resources and reminders; and 3) creating a heightened awareness about AMR; and second, barriers: time pressure (mostly linked to patient volume).

In the paragraph on Delayed prescribing, it is not established whether this strategy inhibits the prescription of antibiotics.

Author Response

I think the work is very well done.

However, I allow myself to make two observations.

Missing reference in paragraph: line 124. A recent qualitative study identified factors affecting the uptake of educational interventions by GPs: first, facilitators: 1) having flexible and relevant learning strategies; 2) having easy to access information, resources and reminders; and 3) creating a heightened awareness about AMR; and second, barriers: time pressure (mostly linked to patient volume).

Reply: We added the missing reference.

In the paragraph on Delayed prescribing, it is not established whether this strategy inhibits the prescription of antibiotics.

Reply:  We cite a systematic review (Spurling et al, 2017) on the effect of delayed prescribing for respiratory infections on antibiotic use. The results show that delayed antibiotics for people with acute respiratory infection reduced antibiotic use compared to immediate antibiotics. We clarified this finding in the manuscript.

Reviewer 4 Report

I want to congratulate the authors for a very interesting review  regarding factors influencing antimicrobial prescription.

My concerns are related to:

- all over the document citations are positioned after the end of sentence .(_) and my personal opinion is that citation should be before the end of sentence (_). as the information cited is within the sentence are should be referred correctly.

- the generality of the review is very high. I would personally enjoy more details about cut-off levels of C reactive protein and procalcitonin to indicate bacterial infection (line 180-192) , also for different guidelines (line 195-199), number of years of experience of GP (line 205), age of GP (221) associated with antimicrobial prescription. Could you be more specific about the data collected?

- provider feedback - how was data obtained? questionnaire?, if so, please give details about the number of individuals assessed and questioned. 

- quality circles, education of the public also should be more detailed

The Conclusion section is quite vague. It contains more ambiguity then the whole document. I would recommend shorted conclusions and maybe stated in a positive way. What was the aim of this article? to identify factors that reduce antibiotic consumption (what exactly works, so other people can apply).

The article starts with data about antimicrobial resistance in introduction section and abstract, but later in the review the antimicrobial resistance is not mentioned any more. Either you add data about antibiotic resistance in different countries studies were applied with effective measures (what worked) or you eliminate the antimicrobial resistance part and change introduction and abstract accordingly. 

The Methods section can be improved with a detailed description of reviews and primary studies included within this review (how many studies, where from, the number of subjects maybe, techniques, etc)

Author Response

#4

I want to congratulate the authors for a very interesting review  regarding factors influencing antimicrobial prescription.

My concerns are related to:

- all over the document citations are positioned after the end of sentence .(_) and my personal opinion is that citation should be before the end of sentence (_). as the information cited is within the sentence are should be referred correctly.

We thank the reviewer or the nice comment on our manuscript.

We corrected the position of the references through the text.

- the generality of the review is very high. I would personally enjoy more details about cut-off levels of C reactive protein and procalcitonin to indicate bacterial infection (line 180-192), also for different guidelines (line 195-199), number of years of experience of GP (line 205), age of GP (221) associated with antimicrobial prescription. Could you be more specific about the data collected?

We added specification about the data we present:

·  We provided more detailed information on the biomarkers cut-off levels to guide prescription of antibiotics.

·  We clarified the median number of year of experience of GPs among those who overruled PCT guidance and those who did not.

·  We clarified that these are mainly guidelines on common infections,such as upper respiratoy tract infections.

·  We modified our statement regarding the age of GPs as data are conflicting: “Data on the age of GPs are inconsistent, with some studies showing an association between older age and antibiotics prescription, others not.”

- provider feedback - how was data obtained? questionnaire?, if so, please give details about the number of individuals assessed and questioned. 

Provider feedback is a system-wide strategy. Data are usually obtained from health insurance claims data or electronic health records as stated in the manuscript. Ususally, it includes thousands of physicians. This information was added tot he manuscript.

- quality circles, education of the public also should be more detailed

We provided more details on quality circles and the involvement of pharmacists. We also added information on education of the public.

The Conclusion section is quite vague. It contains more ambiguity then the whole document. I would recommend shorted conclusions and maybe stated in a positive way. What was the aim of this article? to identify factors that reduce antibiotic consumption (what exactly works, so other people can apply).

We shortened the conclusion and summarized factors influencing implementation of antimicrobial stewardship in primary care.

The article starts with data about antimicrobial resistance in introduction section and abstract, but later in the review the antimicrobial resistance is not mentioned any more. Either you add data about antibiotic resistance in different countries studies were applied with effective measures (what worked) or you eliminate the antimicrobial resistance part and change introduction and abstract accordingly. 

We simplified the introduction and the abstract to put less weight on resistance. However, the global increase in antimicrobial resistance is the reason why coutries have developped their national programmes to optimise antibiotic use. The mention of antimicrobial in the introduction of the manuscript serves to set the context and introduce the reason for discussing antimicrobial stewardship activities and factors influencing their implementation. The goal of this review is not to discuss the impact of these interventions on resistance.

The Methods section can be improved with a detailed description of reviews and primary studies included within this review (how many studies, where from, the number of subjects maybe, techniques, etc

In this narrative review, we discuss 30 studies reporting on factors influencing the implementation of antimicrobial stewardship in primary care. We added a table (Table 1), which shows the characteristics of the studies included in our review.

Round 2

Reviewer 4 Report

I want to congratulate the authors for the good work they put up in revising this article.